# Design and Assembly of a Biofactory for (2*S*)-Naringenin Production in *Escherichia coli*: Effects of Oxygen Transfer on Yield and Gene Expression

**DOI:** 10.3390/biom13030565

**Published:** 2023-03-20

**Authors:** Laura E. Parra Daza, Lina Suarez Medina, Albert E. Tafur Rangel, Miguel Fernández-Niño, Luis Alberto Mejía-Manzano, José González-Valdez, Luis H. Reyes, Andrés Fernando González Barrios

**Affiliations:** 1Grupo de Diseño de Productos y Procesos (GDPP), Department of Chemical and Food Engineering, Universidad de Los Andes, Bogotá 110311, Colombia; 2Department of Bioorganic Chemistry, Leibniz-Institute of Plant Biochemistry, 06120 Halle, Germany; 3Tecnológico de Monterrey, School of Engineering and Science, Av. Eugenio Garza Sada 2501 Sur, Monterrey 64849, NL, Mexico

**Keywords:** (2*S*)-naringenin, oxygen, malonyl-CoA, dissolved oxygen

## Abstract

The molecule (2*S*)-naringenin is a scaffold molecule with several nutraceutical properties. Currently, (2*S*)-naringenin is obtained through chemical synthesis and plant isolation. However, these methods have several drawbacks. Thus, heterologous biosynthesis has emerged as a viable alternative to its production. Recently, (2*S*)-naringenin production studies in *Escherichia coli* have used different tools to increase its yield up to 588 mg/L. In this study, we designed and assembled a bio-factory for (2*S*)-naringenin production. Firstly, we used several parametrized algorithms to identify the shortest pathway for producing (2*S*)-naringenin in *E. coli*, selecting the genes phenylalanine ammonia lipase (*pal*), 4-coumarate: CoA ligase (*4cl*), chalcone synthase (*chs*), and chalcone isomerase (*chi*) for the biosynthetic pathway. Then, we evaluated the effect of oxygen transfer on the production of (2*S*)-naringenin at flask (50 mL) and bench (4 L culture) scales. At the flask scale, the agitation rate varied between 50 rpm and 250 rpm. At the bench scale, the dissolved oxygen was kept constant at 5% DO (dissolved oxygen) and 40% DO, obtaining the highest (2*S*)-naringenin titer (3.11 ± 0.14 g/L). Using genome-scale modeling, gene expression analysis (RT-qPCR) of oxygen-sensitive genes was obtained.

## 1. Introduction

The search for novel molecules with potential health benefits has increased with the demand for natural bioproducts. Because of this demand, scientists and researchers are actively searching for natural sources of biologically active molecules, with the intention of discovering novel compounds that have the potential to be utilized in medical treatments. Polyphenols are appealing molecules that have a bioactive effect on human health and have been cataloged as potential candidates for treating a number of diseases. These diseases include cancer, cardiovascular disease, and neurodegenerative disorders [1,2]. These bioactive effects are related to the type of polyphenol and its individual characteristics, which can vary greatly depending on the origin of the source and the extraction technique [3]. Polyphenols are classified into phenolic acids, flavonoids, stilbenes, and lignans [3]. The molecule (2*S*)-naringenin is a flavonoid that is considered a scaffold molecule. It is estimated that 8000 molecules could be obtained; for example, epicatechin, quercetin, and kaempferol can be obtained from (2*S*)-naringenin [3,4]. Several studies have shown that (2*S*)-naringenin possesses antimicrobial, antibacterial, antiviral, anti-inflammatory, and antioxidant activity [3,5,6]. It can also reduce lipid accumulation in the body, have an antiviral effect against SARS-CoV-2 (COVID-19), impede the reproduction of the hepatitis C virus, and inhibit the proliferation of cancer cells [7,8,9,10].

It has been discovered that plants are capable of (2*S*)-naringenin biosynthesis, and this process has been linked to both the maturation process and the defense mechanism [11]. Because of this, there has been a rise in the amount of research conducted over the past few years regarding the production of (2*S*)-naringenin [4,12]. Both chemical synthesis and isolation from plant extracts have been extensively researched as potential methods of producing (2*S*)-naringenin. In chemical synthesis, hazardous chemicals and labor-intensive purification steps are both necessary and common. Current plant extraction protocols cannot be considered environmentally friendly due to the high water and energy demands required, along with the use of organic solvents that can be difficult to dispose of in an environmentally friendly manner [4,13]. To overcome these and other issues related to (2*S*)-naringenin production, heterologous biosynthesis has emerged as an attractive alternative to improving yield via metabolic engineering or fermentation conditions optimization (e.g., temperature, aerobic/anaerobic conditions, and inducer) [4,14,15,16]. So far, the microbial production of (2*S*)-naringenin has been reported in *Escherichia coli* and *Saccharomyces cerevisiae* using phenylpropanoid precursors and biosynthesis components (Table 1). *E. coli* and *S. cerevisiae* have been shown to produce 2*S*-naringenin at titers as high as 588 mg/L and 648.63 mg/L, respectively. These are the highest titers that have been reported [16,17]. Nonetheless, several strategies have been used to optimize (2*S*)-naringenin production, such as the used a cipher of evolutionary design (CiED) to predict gene deletions, and the development a biosensor to respond to dynamical environmental changes [16,18,19]. In order to obtain commercially viable microbial factories, it is necessary to study variables such as biosynthetic pathways and gene selection, fermentation conditions, and microorganism type using a systematic approach.

In this work, we developed an *E. coli* bio-factory to produce (2*S*)-naringenin. We used OptStoic and MinFlux to predict and identify the shortest pathway for obtaining (2*S*)-naringenin [21]. We defined a set of parameters to select the gene pathways. Then, we used the ePathBrick method for pathway assembly [22] to evaluate the effects of oxygen in a flask (250 mL) and bench (4 L) scales on (2*S*)-naringenin production.

## 2. Materials and Methods

### 2.1. Pathway Selection and Sensitivity Analysis

OptStoic and MinFlux [21] were implemented to identify the shortest route for (2*S*)-naringenin production in *E. coli*. OptStoic was used to determine the stoichiometric equation for producing a desired product from a substrate. MinFlux was implemented to reduce the total flux values [21]. The Kyoto Encyclopedia of Genes and Genome (KEGG) was retrieved during Minflux runs [23].

The *E. coli* model, proposed by Tafur Rangel and colleagues [24], was used for the flux balance analysis (FBA). We used FBA to predict and optimize a metabolic network’s steady-state reaction flux distributions [25,26]. To maximize the biomass function while keeping linear equalities constraints in mind, linear optimization was used. COBRA Toolbox 2.0 and Gurobi 8.1 were synchronized with Matlab 2017a. The shadow prices were calculated to obtain the most oxygen-sensitive metabolic fluxes [27].

### 2.2. Heterologous Pathway Construction and Assembly

The following criteria were used to select the genes used in the assembly of the (2*S*)-naringenin pathway in *E. coli*: (i) genes previously expressed in *E. coli*; (ii) enzymes with Michaelis–Menten constants reported within an order of magnitude of ×10^−2^; and (iii) genes carrying mutations shown to improve (2*S*)-naringenin production. We used BRaunschweig ENzyme DAtabase (BRENDA) [28] to obtain the kinetic parameters for different organism genes that fulfilled all the selection criteria.

Genes of the (2*S*)-naringenin pathway were codon-optimized for *E. coli* BL21 (DE3) expression and synthesized by Shine Gene (Shanghai, China). The ePathBrick method was used to build the (2*S*)-naringenin pathway [22]. Plasmids were created in a monocistronic form, with each gene containing its own promoter and terminator. We ensured similar gene transcription levels with this conformation. The restriction enzymes NheI, SalI, and AvrII were used as needed. Finally, two expression vectors were created as follows: pETM6 (pEPACH) assembled the genes *pal* and *chi*, while vector pRSM3 contained the genes *4cl* and *chs* (pR4CCH) (Table 2).

### 2.3. Strains and Media

Lysogeny broth (LB) was used for growth and flavonoid production. Isopropyl-β-D-thiogalactopyranoside (IPTG) was utilized to induce (2*S*)-naringenin production. Ampicillin (100 μg/mL) and kanamycin (50 μg/mL) were added when required. The flavanone standards (2*S*)-naringenin and p-coumaric acid were purchased from Sigma-Aldrich (St. Louis, MO, USA). Restriction enzymes and T4 DNA ligase were purchased from New England Biolabs (Ipswich, MA, USA). The plasmids (pETM6 and pRSM3) were acquired from Addgene (Watertown, MA, USA). Absorbance at 600 nm was measured with a UV/Vis spectrophotometer (Cary 60 UV-Vis—Agilent, Santa Clara, CA, USA) to track cell growth.

### 2.4. 50 mL Shake Flask (2S)-Naringenin Culture

Unless another temperature is specified, cell cultures are grown at 37 °C. We evaluated the effect of orbital shaking speeds (Thermo Scientific, Waltham, MA, USA) (50, 100, 150, 200, and 250 rpm). Cells were cultured at a starting OD_600_ of 0.1. Induction of (2*S*)-naringenin production was performed during the mid-exponential phase of cultivation (OD_600_ between 0.6 to 0.8) by adding 1 mM IPTG. After the induction, the cell cultures were grown at 30 °C, and the (2*S*)-naringenin production was carried out for 24 h.

### 2.5. 4-L Cultures

Production of (2*S*)-naringenin in *E. coli* was performed in a 5L BioFlo/CelliGen 115 (Eppendorf New Brunswick™, Hamburg, Germany). The conditions for naringenin production were as follows: 4 L of culture medium and the percent of dissolved oxygen (DO) were kept constant in the study at 5% and 40%. Cell inoculation was carried out with a 400 mL pre-inoculum taken from a sample at a steady state. The (2*S*)-naringenin induction and production conditions were the same as those implemented in shake flask (2*S*)-naringenin fermentation.

### 2.6. Analytical Procedures

To analyze the extracellular (2*S*)-naringenin production, four samples at 2, 4, 22, and 24 h were collected. Media samples were diluted with ethanol at a 1:1 relation. After vortexing for two minutes, samples were centrifuged at 16,000× *g* for 2 min. The samples were filtered through a 0.22 µm hydrophobic filter. The (2*S*)-naringenin standard was dissolved in ethanol at different concentrations to prepare the calibration curve.

Samples were analyzed by high-performance liquid chromatography (HPLC) using an LC Agilent 1290 infinity equipment (Agilent Scientific Instruments, Santa Clara, CA, USA) and a reverse phase column ZORBAX Eclipse XDB-C18 (5 µm, 4.6 × 150 mm) maintained at 25 °C with 800 µL injection volume. The (2*S*)-naringenin was separated and analyzed by elution with acetonitrile (0.1% *v*/*v* formic acid) and water (0.1% *v*/*v* formic acid) gradient at a flow rate of 1 mL/min under the following conditions: increasing from 10% to 28% acetonitrile during 20 min. From 20 to 25 min, the flow was set to 28% acetonitrile, followed by an increase to 30% acetonitrile until 30 min.

### 2.7. Gene Expression Analysis

Based on the FBA sensitivity analysis results, ten sensitive genes related to oxygen levels and eight genes involved in the production of (2*S*)-naringenin were chosen to be studied using Reverse Transcription Quantitative Polymerase Chain Reaction (RT-qPCR).

Cells for RNA extraction were harvested in 1 mL aliquots by centrifugation for 2 min, 16,000× *g*, at 4 °C, and the cell pellets were frozen and kept at −80 °C until further use. Total RNA was isolated before the induction and 4 h after inducing the (2*S*)-naringenin production. The Monarch Total RNA Miniprep Kit (New England Biolabs, Ipswich, MA, USA) was used according to the manufacturer’s instructions. Then, RNA was treated with DNase I (New England Biolabs, Ipswich, MA, USA) for 15 min following the manufacturer’s protocol. Additionally, cDNA synthesis and qPCR were performed using the Luna Universal One-Step RT-qPCR kit (New England Biolabs, Ipswich, MA, USA). RT-qPCR was performed in Rotor-Gen Q (Qiagen, Hilden, Germany). The reaction was performed in a total reaction mixture volume of 20 µL containing 0.25 µg of the RNA template. The reaction mixture was incubated for 10 min at 50 °C to synthesize the cDNA. Then, it was set for 1 min at 95 °C for Taq activation, followed by 45 cycles of 10 s at 95 °C (denaturation) and 30 s at 60 °C (annealing/extension). Forward and reverse primers were designed for each gene to generate a PCR product between 50–150 bp, and the percentage of GC was 50% (Appendix A). The *rrsA* gene encoding 16S ribosomal RNA was used as a housekeeping gene.

The Delta-Delta CT method was used for the relative quantification of gene expression [29]. The difference between the CT target gene in the test and calibrator samples ($\Delta C_{T}$) was calculated for all the genes. The $\Delta \Delta C_{T}$ was calculated by taking the difference between $\Delta C_{T}$ of a gene with and without induction. The relative expression level was calculated using the formula $2^{-\Delta \Delta C_{T}}$ [29].

### 2.8. Statistical Analysis

At each stage of the project (50 mL shake flask, 4 L culture, and gene expression analysis) we perform one-way analysis of variance (ANOVA) and Tuckey post host test to understand whether aeration has an impact on (2*S*)-naringenin production. We used at 5% of significance level. We used Minitab 19 to perform this statistical analysis.

## 3. Results

### 3.1. Prediction of the Shortest Pathway for Heterologous (2S)-Naringenin Production

We created a KEGG database to predict the shortest pathway for (2*S*)-naringenin biosynthesis [23], including 10,195 reactions and 18,120 metabolites. We chose L-tyrosine as the substrate to simulate the (2*S*)-naringenin production in *E. coli*. Using OptStoic, we observed that 14 oxygen molecules are needed to obtain (2*S*)-naringenin from L-tyrosine (Equation (1)).
(1)$$12\;L-Tyr+14\;O_{2\;}\to \;5\;\left(2S\right)-\mathrm{naringenin}+3\;C_{2}H_{4}O_{2\;}+3\;C_{6}H_{14}N_{4}O_{2}+9\;H_{2}O+9\;CO_{2}$$

We used Equation (1) to implement the minFlux model in *E. coli* to determine the pathway from L-tyrosine. In an optimized pathway, four catalytic steps were required to produce (2*S*)-naringenin (Figure 1). L-tyrosine is converted to p-coumaric acid by the enzyme tyrosine ammonia-lyase (TAL). Then, p-Coumaric acid is converted into coumaroyl-CoA through the 4-coumarate-CoA ligase (4CL). The enzyme chalcone synthase (CHS) catalyzed the stepwise condensation of three malonyl-CoA molecules with one coumaroyl-CoA molecule. The resulting naringenin chalcone is converted to (2*S*)-naringenin by the enzyme chalcone isomerase (CHI). In literature, it has been reported that (2*S*)-naringenin could be obtained from phenylalanine ammonia-lyase (PAL) using 4CL, CHS, and CHI enzymes [30,31].

### 3.2. Selection of Pathway Genes Sequence

We selected each gene following the criteria. For the first step of naringenin synthesis, we searched for genes producing the TAL, PAL, or TAL/PAL enzyme. The gene from *Bambusa oldhamii*/BoPal4 is PAL and has a mutation (F133H) that increased the affinity for L-tyrosine [32]. According to the authors, this may contribute to the pathway to (2*S*)-naringenin production [32]. We identify two options to select a *4cl* sequence: *Solanum lycopersicum* with Q274H mutation [33] and *S. lycopersicum* with Q274H-F269L mutations [33]. We selected the *S. lycopersicum* with Q274H mutant due to reaction selective stabilization and abolition of the inhibition site in (2*S*)-naringenin presence [33]. Continuing with the *chs* sequence, we selected the gene from *Hypericum androsaemum* [34] because of the increased enzyme activity to 4-coumaroyl: CoA ligase. This aspect is reflected in the increased production of naringenin chalcone and, in turn, in the (2*S*)-naringenin production. The *chi* sequence selected comes from the organism *Pueraria montana* var. Lobata [35]; this one was chosen since it is the only one reported in BRENDA. Candidates for each gene are shown in Table 3.

### 3.3. Selection of Genes Sensitive to Oxygen Variation

To select the genes sensitive to oxygen variation, we performed a Phenotypic Plane Phase analysis (PhPPs) to determine the shadow prices in each phase, obtaining six phases in our PhPPs (Appendix A). Our model displays 2063 metabolites in extracellular, periplasm, or cytoplasm compartments. We evaluated each metabolite in each phase on PhPPs and selected those metabolites with shadow price values four-fold above the average in each phase. Then, we assessed the reaction flux for each metabolite in each plane, selecting those metabolites based on gene-to-protein relations and displaying associated reactions with non-zero flux values (Table 4).

### 3.4. (2S)-Naringenin Production at a Laboratory Scale

We evaluated how different aeration conditions could affect (2*S*)-naringenin production during fermentation (Figure 2) by analyzing extracellular (2*S*)-naringenin production at different time frames (2, 4, 22, and 24 h). The highest (2*S*)-naringenin titer was obtained at 50 rpm for 24 h (Figure 2A) with the lowest p-Coumaric acid production (Figure 2B) and biomass (Figure 2C). Between 4 h and 22 h, heterologous (2*S*)-naringenin production increased at 50 rpm, 100 rpm, 200 rpm, and 250 rpm, while it decreased at 150 rpm (Figure 2A). Between 22 h and 24 h, two (2*S*)-naringenin profiles were obtained. At low aeration conditions (50 rpm and 100 rpm), (2*S*)-naringenin production increases, and high aeration conditions, (2*S*)-naringenin production decreases (Figure 2A). The production of p-Coumaric acid increases between 4 h and 24 h (Figure 2B).

To corroborate the results, we perform an ANOVA at 5% of significance level for each time. In the four stages of the experiment, we found that in each hour at least one of agitation rate conditions are different for (2*S*)-naringenin production.

### 3.5. (2S)-Naringenin Production at Pilot Scale

Based on the obtained (2*S*)-naringenin production in the shake flask fermentation (Figure 2), we decided to produce (2*S*)-naringenin in a 4 L bioreactor culture under 5% and 40% dissolved oxygen (DO) conditions (Figure 3). Maximum (2*S*)-naringenin production for 5% DO is obtained at 4 h (2.95 ± 0.13 g/L), and 40% DO is obtained at 2 h (3.11 ± 0.14 g/L) (Figure 3A). There is a decrease (2*S*)-naringenin production over time at 5% DO, a reduction of 47.08% from 22 h to 24 h (Figure 3A). The production of p-Coumaric acid increases over time in both conditions (Figure 3B). We perform an ANOVA to verify whether DO has an impact on (2*S*)-naringenin production. Only at 24 h did we obtain that DO conditions are different at a 5% significance level.

### 3.6. Gene Expression Analysis at Different Oxygen Transfer Conditions

RT-qPCR is one of the most-used methods for determining the abundance of a transcript [38]. It is possible to compare the gene expression change under a specific condition. Samples for RT-qPCR were taken four hours after inducing (2*S*)-naringenin production at 5% and 40% dissolved oxygen to evaluate the expression at the 4 L scale and the effect of oxygen transfer on (2*S*)-naringenin production at laboratory scale (Figure 4). On the one hand, fatty acids, pyrimidine metabolism, and division cell genes display a higher expression at 40% of dissolved oxygen (*fadE*, *fadB*, *fadI*, *testB*, *pyrD*, *mraY,* and *murG* are upregulated). On the other hand, the expression of some genes related to glycolysis, TCA, and (2*S*)-naringenin synthesis is higher at low oxygen concentrations (*frdA*, *cyoD*, *paI*, and *accA*). According to ANOVA, the genes *fadB*, *fadI*, *pyrD*, *nuoA*, *cyoD*, *IpdA*, *pal*, *chs* and *accA* have a different relative expression at 5% of significance level.

## 4. Discussion

Pathway design, bio-factory selection, and scale-up are all required for the heterologous microbial synthesis of metabolites. In this work, we combined the three steps for creating a valuable molecule: (2*S*)-naringenin. We found that stoichiometric-based approaches (OptStoic and MinFlux) that facilitate gene selection must consider the synthesis platform and kinetics. By considering these factors, we reported the highest naringenin concentration. Furthermore, we discovered that the effects of oxygen transfer depend on the fermentation volume, which should be considered during scale-up. The PhPP analysis summarizes previous research on malonyl coenzyme A’s effects on flavanone synthesis, as well as the role of the redox state inside the cell, which could explain the role of oxygen during fermentation. On a flask scale, we varied the agitation rate to evaluate the effect of oxygen on (2*S*)-naringenin production. At a high oxygenation rate, (2*S*)-naringenin levels are found to decrease over time (Figure 2). Then, at a low oxygenation rate, a higher (2*S*)-naringenin production was obtained. This result could be explained by the fact that *E. coli* uses (2*S*)-naringenin for other metabolic pathways because high oxygen transfer favors the tricarboxylic acid cycle as an energy source, affecting ACCOAC flux and, thus, (2*S*)-naringenin production, which could allow p-Coumaric acid accumulation (Figure 2B). Previous studies have shown an accumulation of p-coumaric acid due to decreased (2*S*)-naringenin synthesis [16] due to the availability of malonyl-CoA when the organism reaches the stationary phase.

As a point of comparison with (2*S*)-naringenin production at a pilot scale, acetyl-CoA carboxylase has been overexpressed, and 3 mM of L-tyrosine was added to the culture, to obtain 60 mg/mL of (2*S*)-naringenin [14]. Cipher of evolutionary design model (CiED) has been used to evaluate possible genetic perturbations that modify the phenotype, obtaining 215 mg/mL of (2*S*)-naringenin [18]. Modular optimization and added sodium malonate dibasic to the medium has been used to obtained 100.64 mg/mL of (2*S*)-naringenin [4]. Designed a growth-coupled biosensor has been used to regulated p-Coumaric acid and malonyl-CoA to optimized (2*S*)-naringenin production, was obtained 215.4 mg/mL of (2*S*)-naringenin [19]. Our study obtained the lowest (2*S*)-naringenin production at a laboratory scale. However, we did not perform any genetic modification, add any reagent to the medium (e.g., sodium malonate dibasic), or perform a biosensor for (2*S*)-naringenin production as in the studies previously mentioned. In this study, we searched for a set of parameters to select pathway genes and evaluate the effect of oxygen on (2*S*)-naringenin production. It is possible that implementing some of these strategies would increase (2*S*)-naringenin production.

Regarding (2*S*)-naringenin production in bench conditions, a maximum of 3.11 ± 0.14 g/L was obtained at 40% dissolved oxygen after 2 h of induction (Figure 3A). This is the highest (2*S*)-naringenin production reported to our knowledge. It has been evaluated the culture’s medium components, temperature, and pH to enhance (2*S*)-naringenin production in a 5 L bioreactor, and it was discovered that these variables improve its production, resulting in 588 mg/L [19]. Furthermore, they reported a decrease in the concentration of (2*S*)-naringenin, which agrees with our findings, as future research should investigate why *E. coli* uses naringenin in its cellular metabolism and evaluate alternatives to avoid this. We supposed that by assessing the effect of the temperature and pH of the culture, we could obtain a better (2*S*)-naringenin production.

Finally, we performed RT-qPCR on the sensitive genes obtained by phenotype phase plane analysis to understand the impact of oxygen transfer on naringenin production. At 40% DO, the relative expression of the *chs* and *chi* genes are favored, while at 5% DO, the relative expression of the *accA* gene is favored (Figure 4E). At 5% DO, the relative expression of the *chi* is low. This gene is responsible for producing the CHI enzyme, which converts (2*S*)-naringenin chalcone to (2*S*)-naringenin. At 40% DO, there is a low relative expression of the *accA* gene in charge of the reaction that converts acetyl-CoA to malonyl-CoA. Since there are not enough malonyl-CoA molecules, (2*S*)-naringenin cannot be produced. These results show that oxygen favors the expression of specific pathway genes, indicating why the (2*S*)-naringenin production in both conditions is similar. We supposed that a greater yield could be obtained by varying the oxygen availability along the (2*S*)-naringenin production.

## Figures and Tables

**Figure 1 biomolecules-13-00565-f001:**
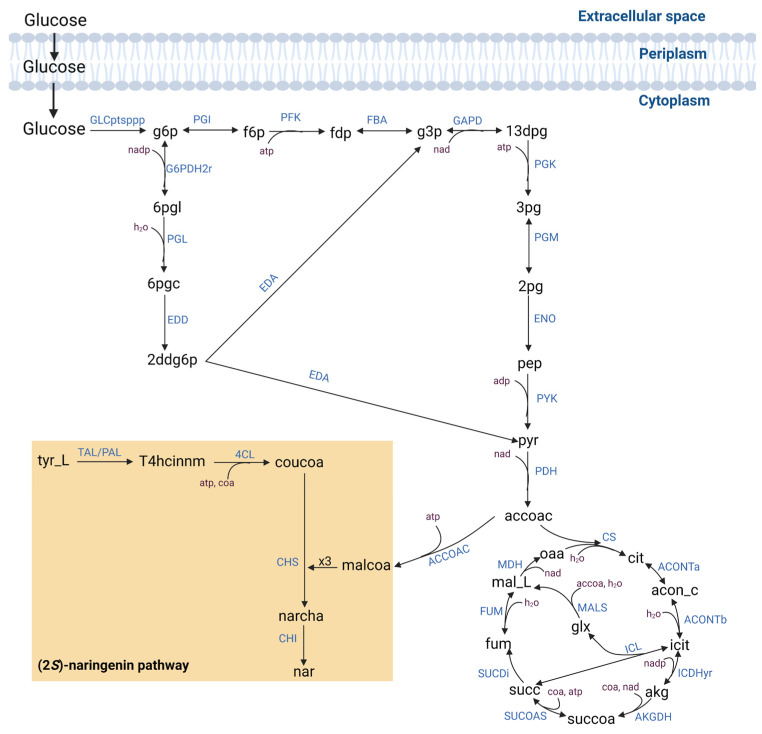
Heterologous pathway for biosynthesis of (2*S*)-naringenin in *Escherichia coli*. Metabolites: pep: phosphoenolpyruvate, pyr: pyruvate, accoac: acetyl-CoA, malcoa: malonyl-CoA, cit: citrate, tyr_L: L-tyrosine, T4hcinnm: p-coumaric acid, coucoa: coumaroyl-CoA, narcha: naringenin chalcone, nar: (2*S*)-naringenin. Reaction: PYK: pyruvate kinase, PDH: pyruvate dehydrogenase, ACCOAC: acetyl-CoA carboxylase, CS: citrate synthase TAL: tyrosine ammonia-lyase, PAL: phenylalanine ammonia-lyase, 4CL: 4-coumarate-CoA ligase, CHS: naringenin chalcone synthase, and CHI: chalcone isomerase. Metabolites are shown in black letters. Reactions are shown in blue letters, and the optimized route for (2*S*)-naringenin production is presented in the orange box.

**Figure 2 biomolecules-13-00565-f002:**
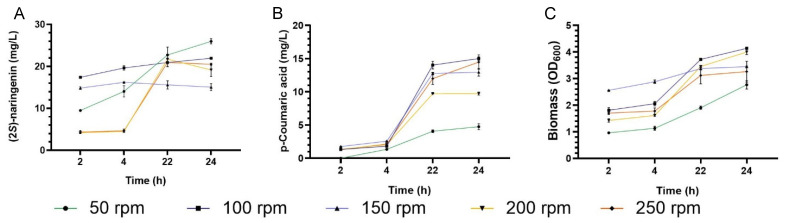
The (2*S*)-Naringenin heterologous production in different aeration conditions at a laboratory scale. (**A**) The (2*S*)-naringenin heterologous production in *E. coli*. (**B**) The p-Coumaric acid heterologous production in *E. coli*. (**C**) Cell growth of *E. coli*.

**Figure 3 biomolecules-13-00565-f003:**
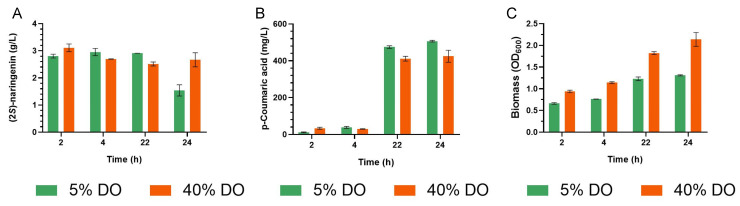
The (2*S*)-Naringenin heterologous production at scale pilot. (**A**) The (2*S*)-naringenin heterologous production in *E. coli*. (**B**) The p-Coumaric acid heterologous production in *E. coli*. (**C**) Cell growth of *E. coli*. The green bars represent the 5% oxygen dissolved condition. The orange bars the 40% oxygen dissolved condition.

**Figure 4 biomolecules-13-00565-f004:**
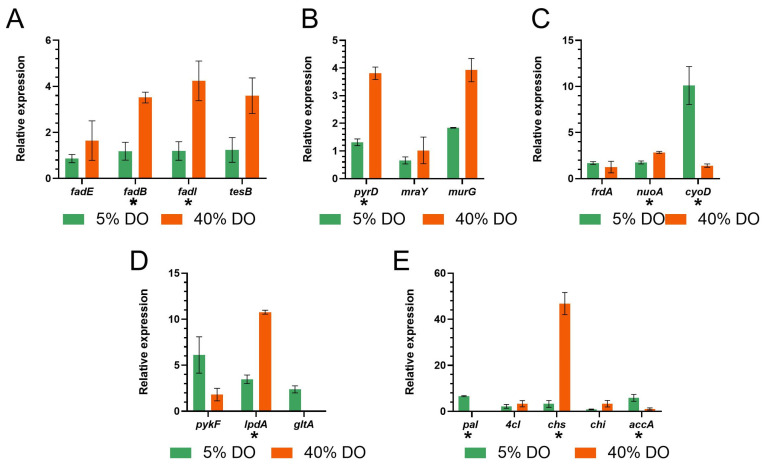
Relative expression of genes sensitive to oxygen flux variation. (**A**) Fatty acids genes. (**B**) Pyrimidine metabolism and division cell genes. (**C**) Electron transport genes. (**D**) Glycolysis and TCA genes. (**E**) (2*S*)-naringenin pathway genes. Statistical analysis was performed with 5% of significance level. Genes that have a * means that are oxygen sensitive genes.

**Table 1 biomolecules-13-00565-t001:** Heterologous production of (2*S*)-naringenin in *Escherichia coli* and *Saccharomyces cerevisiae* strains.

Microorganism	Biosynthesis Genes	Titer (mg/L)	Reference
*E. coli* BLR (DE3)	*pal*, *4cl*, *chs*, *chi and acc*	57	[14]
*E. coli* BL21 Star^TM^	*4cl*, *chs*, *chi*, *acc*, *pgk*, *pdh*	474	[15]
*E. coli* rpoA14^R^	*tal*, *4cl*, *chi*, *chs*	84	[20]
*S. cerevisiae* CEN.PK2-1c	*pal*, *c4h*, *cpr*, *4cl*, *chs*, *chi*, *tal*	109	[12]
*E. coli* BL21 (DE3)	*4cl*, *chs*, *chi*	588	[19]
*S. cerevisiae* CEN.PK2-1D	*tal*, *4cl*, *chs*, *chi*, *aroL*, *ARO7fb*, *ARO4fbr*	648.93	[17]

**Table 2 biomolecules-13-00565-t002:** Strains and plasmids used in this study.

Strain	Genotype	
*Escherichia coli* BL21(DE3)	F^−^ *omp*T *hsd*S_B_ (r_B_^−^, m_B_^−^) *gal dcm* (DE3)	
*Escherichia coli* DH5α	F–φ80lacZΔM15 Δ(*lacZYA*-*argF*)U169 *recA*1 *endA*1 *hsdR*17(rK^−^, mK^+^) *phoA* supE44 λ–thi-1 *gyrA*96 *relA*1	
Plasmid	Description	Source of reference
pETM6	T7 promoter, ampicillin resistance	Addgene
pRSM3	T7 promoter, kanamycin resistance	Addgene
pEPACH	pETM6 carrying *pal* and *chi* genes	This study
pR4CCH	pRSM3 carrying *4cl* and *chs* genes	This study

**Table 3 biomolecules-13-00565-t003:** List of possible organisms for each gene found for the synthesis of (2*S*)-naringenin.

Organism	Km (mM)	Mutation Presence	References
pal gene			
*B. oldhamii*/BoPAL1	0.230	Yes	[32]
*B. oldhamii*/BoPAL2	0.333	Yes	[32]
*B. oldhamii*/BoPAL4	0.097	Yes	[32]
*R. toruloides*	0.180	No	[36]
4cl gene			
*S. lycopersicum* Q274H	0.179	Yes	[33]
*S. lycopersicum* Q274H-F269L	0.036	Yes	[33]
chs gene			
*H. androsaemum*	0.023	Yes	[34]
*M. sativa*	0.023	Yes	[37]
chi gene			
*H. androsaemum*	0.023	Yes	[35]

**Table 4 biomolecules-13-00565-t004:** List of genes sensitive to oxygen variation obtained through shadow prices analysis.

Descriptive Name	Function
(*fadE*) Acyl-coenzyme A dehydrogenase	Fatty acids
(*fadB*) Fatty acid oxidation complex, alpha subunit	Fatty acids
(*tesB*) Acyl-CoA thioesterase 2	Fatty acids
(*fadI*) 3-ketoacyl-CoA thiolase	Fatty acids
(pyrD) Dihydroorotate dehydrogenase	Pyrimide metabolism
(*frdA*) Fumarate reductase flavoprotein subunit	Electron transport gene
(*nuoA*) NADH-quinone oxidoreductase subunit A	Electron transport gene
(*cyoD*) Cytochrome bo(3) ubiquinol oxidase subunit 4	Electron transport gene
(*mraY*) phospho-N-acetylmuramoyl-pentapeptide-transferase	Division cell
(murG) UDP-N-acetylglucosamine-acetylglucosamine transferase	Division cell

## Data Availability

Not applicable.

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
