# Peer review of "Design and Assembly of a Biofactory for (2S)-Naringenin Production in Escherichia coli: Effects of Oxygen Transfer on Yield and Gene Expression"

_biomolecules, 2023, doi:10.3390/biom13030565_

Round 1

Reviewer 1 Report

The manuscript entitled "Design and assembly of a microbial bio-factory for the heterologous expression of (2S)-naringenin in Escherichia coli" describes the development of a bioreactor procedure for the production of (2S)-naringenine using as host E. coli. A serious effort has been made to investigate the role of dissolved oxygene during the process. It is a very interesting work and its novelty is based on the product rather than the overall methodology, but there are a couple of things I would like to discuss.

Major concerns

In all figures there are error bars but nowhere a statistically significance of the observed differences is shown. How many times did the experiments be repeated? Did a statistical analysis be performed?

Minor concerns

There is the increased biomass in the higher value of D.O. which is not accompanied by a similar increment on (2S)-naringenine production. This result clearly shows that acetic acid is driven to citric acid cycle and subsequent biomass development rather to (2S)-naringenine synthesis. There is a possibility, that the use of a citric cycle inhibitor (aconitase inhibitor?) during the process will act beneficial on this. Is it possible this manuscript to be enriched with this additional experiment?

Author Response

Response to reviewers:

The authors wish to thank the reviewers and the chief editor for their insightful and constructive comments on the manuscript. This review process has unquestionably enhanced the quality of our analysis and overall work. The original comments of the reviewers are shown in bold below. Changes to the manuscript are explained and highlighted in the document. We would like to call attention to the fact that the study's title was slightly modified to better reflect its purpose.

Reviewer 1

The manuscript entitled "Design and assembly of a microbial bio-factory for the heterologous expression of (2S)-naringenin in Escherichia coli" describes the development of a bioreactor procedure for the production of (2S)-naringenine using as host E. coli. A serious effort has been made to investigate the role of dissolved oxygene during the process. It is a very interesting work and its novelty is based on the product rather than the overall methodology, but there are a couple of things I would like to discuss.

Major concerns

  • In all figures there are error bars but nowhere a statistically significance of the observeddifferences is How many times did the experiments be repeated? Did a statistical analysisbe performed?

We value the reviewer's feedback. We repeated each experiment three times. We included statistical analysis in every required section of this manuscript.

Minor concerns

  • There is the increased biomass in the higher value of D.O., which is not accompanied by a similar increment on (2S)-naringenine production. This result clearly shows that acetic acid is driven to citric acid cycle and subsequent biomass development rather to (2S)-naringenine synthesis. There is a possibility that the use of a citric cycle inhibitor (aconitase inhibitor?) during the process will actbeneficial on Is it possible this manuscript to be enriched with this additional experiment?

We appreciate your feedback. We are unable to conduct additional experiments because the grant that funded this project has expired. In addition, we lack the personnel necessary to continue this project. However, your suggestion is quite intriguing and will be incorporated into a future contribution.

Reviewer 2 Report

The importance of (2S)-naringenin in food, medicine and other areas cannot be overstated. The authors have based their reported findings on the rearrangement and pathway optimisation of the biological components involved in its biosynthesis, resulting in high yields of g/L. The content of the manuscript is consistent with the main theme of the journal and the following bugs should be revised before it is considered for publication.

1.      In line 95,ePathBrickshould be PathBrick”

2.      The chart in the article is even more modified and beautified.

3.      In line 374,389,Species names in Latin need to be italicized.

4.      2S-naringenin, not 2S-naringenin

5.      The reference format is not the one prescribed by the MDPI journal, please correct it.

Author Response

Response to reviewers:

The authors wish to thank the reviewers and the chief editor for their insightful and constructive comments on the manuscript. This review process has unquestionably enhanced the quality of our analysis and overall work. The original comments of the reviewers are shown in bold below. Changes to the manuscript are explained and highlighted in the document. We would like to call attention to the fact that the study's title was slightly modified to better reflect its purpose.

Reviewer 2

The importance of (2S)-naringenin in food, medicine and other areas cannot be overstated. The authors have based their reported findings on the rearrangement and pathway optimisation of the biological components involved in its biosynthesis, resulting in high yields of g/L. The content of the manuscript is consistent withthe main theme of the journal and the following bugs should be revised before it is considered forpublication.

  • In line 95,“ePathBrick”should be “PathBrick”

Your suggestion is appreciated. According to the authors who created this assembly method, however, the correct name is "ePathBrick" (https://pubs.acs.org/doi/10.1021/sb300016b). Perhaps it is a revised version of "PathBrick."

  • In line 374,389,Species names in Latin need to be

Thank you! We edited and italicized per your request. Hopefully, we've located everyone.

  • 2S-naringenin, not 2S-naringenin

We appreciate your comment. We corrected the inconsistency in the manuscript.

  • The reference format is not the one prescribed by the MDPI journal, please correct

We appreciate your comment. We hope that this time we have it right.

Reviewer 3 Report

In this manuscript, Daza, et al., first built an improved biosynthesis pathway for (2S)-naringenin in E. coli, using tyrosine as the precursor. Then, they optimized this pathway by investigating the naringenin titer with various parameters, such as aeration conditions and dissolved oxygen (DO). The authors further analyzed the expression of several metabolic genes using RT-qPCR.

Research on the heterologous biosynthesis pathway is an interesting topic in the field of metabolic engineering, which usually shows many pharmaceutical and industrial applications. Previously, the biosynthesis pathways of 2S-naringenin have been established in E. coli and yeast. Notably, similar pathways using L-tyrosine as a precursor and the same set of genes (TAL, 4CL, CHS, CHI) were published before (Näring, Gérard, Work & Stress 20.4 (2006): 303-315.), with several follow-up investigations (such as, Wu, Junjun, et al. Applied and environmental microbiology (2014); Ganesan, Vijaydev, et al. Synthetic and systems biotechnology (2017); Dunstan, Mark S., et al. Synthetic Biology (2020)). These previous studies using L-tyrosine as a precursor should be mentioned in the introduction section. In this manuscript, the authors tried optimizing the pathway enzyme components using their predictable model, and achieved the highest titer of ~3 g/L after their test using various parameters. The authors did careful investigations and presented their results clearly. The manuscript was also well-written. All my comments are minor and hope to be addressed by the authors, to make this manuscript better.

1. "TAL" or "PAL"

Clearly, the authors used TAL, tyrosine ammonia-lyase, as mentioned in line 184. PAL, standing for phenylalanine ammonia-lyase, was used several times. This is clearly an error and should be fixed. For example, Figure 1 and its legend, section of "selection of pathway genes sequence." As a reviewer, I understand that the enzyme may have both TAL and PAL activities, which may even switch based on their sequence mutations or variants. But please explain this more clearly and explicitly for general readers.

2. The reference for (Hsieh et al., 2010) (line 203) is missing from the references.

 Perhaps this is the article that was meant to be referenced - https://www.mdpi.com/2073-4344/11/11/1263 ??

Also, line 203, the specific mutation found in pal gene, is not mentioned.

3. The broken columns in Figure 3 and 4 should be avoided.

For example, in Figure 3, I understand that there is a dramatic difference between the coumaric acid concentration at the time of 4 and 22 h. But I also strongly believe the broken column is illegal, because this may "hide" and "mislead" this kind of dramatic difference. Please fix this.

4. No significance test was used.

In Figure 3 and 4, the authors tried to state that the 5% and 40% DO obviously affect the production of 2S-naringenenin, and the expression of several key metabolic genes. But none significance test was included. Please consider including some tests and labeling these columns with the corresponding p-value.

5. Many references were cited in the main article but missed in the "References" section.

For example, line 49, Xu et al, Wu et al,... and citations in Table 1, table 3. Please go through the manuscript again carefully and make sure all these references have been listed. Several are critical, about the previous studies of 2S-naringenenin biosynthesis in E. coli.

Author Response

Response to reviewers:

The authors wish to thank the reviewers and the chief editor for their insightful and constructive comments on the manuscript. This review process has unquestionably enhanced the quality of our analysis and overall work. The original comments of the reviewers are shown in bold below. Changes to the manuscript are explained and highlighted in the document. We would like to call attention to the fact that the study's title was slightly modified to better reflect its purpose.

Reviewer 3

In this manuscript, Daza, et al., first built an improved biosynthesis pathway for

(2S)-naringenin in E. coli, using tyrosine as the precursor. Then, they optimized this pathway by investigatingthe naringenin titer with various parameters, such as aeration conditions and dissolved oxygen (DO). The authors further analyzed the expression of several metabolic genes using RT-qPCR.

Research on the heterologous biosynthesis pathway is an interesting topic in the field of metabolic engineering, which usually shows many pharmaceutical and industrial applications. Previously, thebiosynthesis pathways of 2S-naringenin have been established in E. coli and yeast. Notably, similarpathways using L-tyrosine as a precursor and the same set of genes (TAL, 4CL, CHS, CHI) were publishedbefore (Näring, Gérard, Work & Stress

20.4 (2006): 303-315.), with several follow-up investigations (such as, Wu, Junjun, et al. Applied and environmental microbiology (2014); Ganesan, Vijaydev, et al. Synthetic and systems biotechnology (2017); Dunstan, Mark S., et al. Synthetic Biology (2020)). These previous studies using L-tyrosine as a precursor should be mentioned in the introduction section. In this manuscript, the authors tried optimizing the pathway enzyme components using their predictable model, and achieved the highest titer of ~3 g/L aftertheir test using various parameters. The authors did careful investigations and presented their resultsclearly. The manuscript was also well-written. All my comments are minor and hope to be addressed by theauthors, to make this manuscript better.

  • "TAL" or "PAL" Clearly, the authors used TAL, tyrosine ammonia-lyase, as mentioned in line PAL, standing for phenylalanine ammonia-lyase, was used several times. This is clearly an error and should be fixed. For example, Figure 1 and its legend, section of "selection of pathway genessequence." As a reviewer, I understand that the enzyme may have both TAL and PAL activities, whichmay even switch based on their sequence mutations or variants. But please explain this more clearly and explicitly for general readers.

Thank you for your insightful comments. It is possible to produce (2S)-naringenin using either the TAL or PAL enzyme. This is emphasized in the revised version, and the confusion with TAL and PAL should be resolved.

  • The reference for (Hsieh et al., 2010) (line 203) is missing from the references. Perhaps this is the article that was meant to be referenced - https://www.mdpi.com/2073-4344/11/11/1263) Also, line203, the specific mutation found in pal gene, is not

In this new version, we've added the references you suggested and fixed the one that was missing.

  • The broken columns in Figure 3 and 4 should be avoided. For example, in Figure 3, I understand that there is a dramatic difference between the coumaric acid concentration at the time of 4 and 22 But I also strongly believe the broken column is illegal, because this may "hide" and "mislead" this kind of dramatic difference. Please fix this.

We appreciate your suggestion. Our goal was never to conceal a value; rather, it was to improve the visualization. In either case, figures 3 and 4 were modified in this revised version.

  • No significance test was used. In Figure 3 and 4, the authors tried to state that the 5% and 40% DO obviously affect the production of 2S-naringenenin, and the expression of several key metabolic But none significance test was included. Please consider including some tests and labeling these columns with the corresponding p-value

This revised version includes a statement regarding the statistical analysis.

  • Many references were cited in the main article but missed in the "References" For example, line 49, Xu et al, Wu et al,... and citations in Table 1, table 3. Please go through themanuscript again carefully and make sure all these references have been listed. Several are critical, about the previous studies of 2S-naringenenin biosynthesis in E. coli.

The references that were missing were added. We appreciate your pointing it out.

Round 2

Reviewer 1 Report

The authors corected/added all the necessary information so, i believe the manuscript should be accepted for publication